# Study on the Modeling and Compensation Method of Pose Error Analysis for the Fracture Reduction Robot

**DOI:** 10.3390/mi13081186

**Published:** 2022-07-27

**Authors:** Minghe Liu, Jian Li, Hao Sun, Xin Guo, Bokai Xuan, Lifang Ma, Yuexuan Xu, Tianyi Ma, Qingsong Ding, Baichuan An

**Affiliations:** 1School of Artificial Intelligence and Data Science and Engineering Research Center of Intelligent Rehabilitation Device and Detection Technology, Ministry of Education, Hebei University of Technology, Tianjin 300130, China; hblgdxlmh@163.com (M.L.); gxhebut@aliyun.com (X.G.); xuanbokai@126.com (B.X.); 17302210909@163.com (Y.X.); 202022801015@stu.edu.cn (T.M.); 15900247220@163.com (Q.D.); anbaichuan0719@163.com (B.A.); 2School of Automation, Beijing University of Posts and Telecommunications, Beijing 100876, China; 3Beijing Key Laboratory of Rehabilitation Technical Aids for Old-Age Disability and Key Laboratory of Neuro-Functional Information and Rehabilitation Engineering of the Ministry of Civil Affairs, National Research Center for Rehabilitation Technical Aids, Beijing 100176, China; malifang1105@163.com

**Keywords:** fracture reduction robot, error model, whale optimization algorithm, differential evolution algorithm, opposition-based learning, error compensation

## Abstract

Background: In the process of fracture reduction, there are some errors between the actual trajectory and the ideal trajectory due to mechanism errors, which would affect the smooth operation of fracture reduction. To this end, based on self-developed parallel mechanism fracture reduction robot (FRR), a novel method to reduce the pose errors of FRR is proposed. Methods: Firstly, this paper analyzed the pose errors, and built the model of the robot pose errors. Secondly, mechanism errors of FRR were converted into drive bar parameter’s errors, and the influence of each drive bar parameter on the robot pose error were analyzed. Thirdly, combining with Cauchy opposition-based learning and differential evolution algorithm (DE), an improved whale optimization algorithm (CRLWOA-DE) is proposed to compensate the end-effector’s pose errors, which could improve the speed and accuracy of fracture reduction, respectively. Results: The iterative accuracy of CRLWOA-DE is improved by 50.74%, and the optimization speed is improved by 22.62% compared with the whale optimization algorithm (WOA). Meanwhile, compared with particle swarm optimization (PSO) and ant colony optimization (ACO), CRLWOA-DE is proved to be more accurate. Furthermore, SimMechanics in the software of MATLAB was used to reconstruct the fracture reduction robot, and it was verified that the actual motion trajectory of the CRLWOA-DE optimized kinematic stage showed a significant reduction in error in both the x-axis and z-axis directions compared to the desired motion trajectory. Conclusions: This study revealed that the error compensation in FRR reset process had been realized, and the CRLWOA-DE method could be used for reducing the pose error of the fracture reduction robot, which has some significance for the bone fracture and deformity correction.

## 1. Introduction

With the continuous development of robotics, parallel robots are increasingly used in military and medical applications due to their compact structure, high stiffness, high load-bearing capacity, better isotropy, and small working space [1]. FRR-assisted fracture repositioning with a parallel mechanism is a feasible and accurate method of repositioning that is more minimally invasive in operation, reduces the operator’s physical effort in manual pulling and repositioning, and minimizes X-ray radiation exposure to patients and medical personnel [2,3]. Accuracy and precision are important performance indicators in the FRR resetting process, which directly affects the success of fracture resetting surgery [4,5]. Due to the complex parallel structure of the resetting robot, there are structural errors, such as hinge mounting and assembly [6,7], and there are dynamic errors arising from non-linear factors in the resetting process, which can lead to reduced precision in the resetting process of the resetting robot and have a significant impact. This can lead to reduced accuracy during repositioning, which can have a significant impact on the repositioning procedure. Therefore, there is an urgent need to compensate for the errors of the resetting robot, to eliminate or reduce the end position errors of the fracture resetting robot, and to improve the accuracy of the resetting process [8,9]. This leads to more accurate fracture repositioning [10,11].

Fortunately, a number of methods for compensating for positional errors at the end of a parallel mechanism have attracted the interest of researchers. Firstly, the fracture repositioning robot posture error modelling of the parallel mechanism is mainly performed using the D–H transformation matrix and closed-loop vector method. Lee Sungcheul [12] used a combination of linear uniform transformation matrix and D–H method to build the error model, but the D–H matrix method is not easy to obtain obvious expressions, so this paper uses the closed-loop vector to complete the error modeling. This paper uses a closed-loop vector to complete the error model. The structural error parameter compensation values of the mechanism are obtained by an optimization algorithm. To date, new intelligent optimization algorithms have been developed to search for optimal solutions, including genetic algorithms, neural networks, particle swarm algorithms [13], ant colony algorithms [14], etc. G. Gungor [15] and others have performed online estimations of errors and compensation of errors based on least squares for parallel robots. However, the least squares algorithm is extremely sensitive to the effects of noise and tends to cause unstable iterative results. Angelidis A and Vosniakos G C. [16] and Nguyen [17] used artificial neural nets to measure and compensate for the end-of-execution errors of industrial robots, respectively. According to Wang Ruizhou [18] the use of the multi-objective particle swarm optimization algorithm was proposed to optimize the parameters and, thus, improve the accuracy of the parallel mechanism motion. In [19], different pseudo-random number assignment strategies are introduced to study the effect of controlled randomness on the search scheme of the particle swarm optimization algorithm; however, the particle swarm algorithm is prone to premature convergence and was shown not to be globally convergent. Based on this, this paper introduces the whale optimization algorithm [20]. The whale optimization algorithm has the advantages of a simple process and fast convergence, but the whale optimization algorithm, as a new population intelligence optimization algorithm, still has some shortcomings. In [21], a chaotic feedback adaptive whale optimization algorithm was proposed for the disadvantage of low accuracy in finding the best complex function optimization problems. In [22], a chaotic search strategy-based whale optimization algorithm (CWOA) is proposed, which addresses the problem that exploration and exploitation capabilities are difficult to coordinate and easily fall into a local optimum. In [23], the Lévy flying whale optimization algorithm is used to improve the convergence of the algorithm. An adaptive decision operator-based whale optimization algorithm (IWOA) was proposed in [24] to improve the convergence speed of the algorithm. Scholars have made many improvements and have now achieved better experimental results, making WOA optimization relatively mature. However, the whale optimization algorithm still does not fully solve the problem of global search capability and convergence speed optimization, and is prone to fall into the problem of local optimality. Based on this [25], a chaotic whale optimization algorithm incorporating differential evolution was proposed to improve the performance of the algorithm, and, in [26], a differential evolution (M-WODE) algorithm based on the multi-objective whale optimization algorithm was proposed to ensure the diversity of solutions and enhance the local search ability of M-WODE. In short, the optimized whale optimization algorithm is widely used, but it has not been applied for the end position error compensation of parallel mechanisms, especially for fracture resetting robots with parallel mechanisms.

This paper analyzes the research of recent years and combines different improvement methods. A differential evolution-based Cauchy opposition-based learning whale optimizations algorithm is proposed to compensate for the end-position error of the FRR. This paper analyzes the mechanism error existing in the FRR of the parallel mechanism and establishes its error model, analyzes the influence of the drive rod parameters on the end-position error, and uses the improved whale optimization algorithm (CRLWOA-DE) to find the optimal drive rod parameters to compensate the dynamic platform end-position error of the FRR. Finally, simulation experiments are used to verify the superiority of the algorithm for error compensation in the FRR reset process.

## 2. Structural Design of the Fracture Reduction Robot

The FRR system consists of a controller, sensor, servo motor, parallel 6-DOF robot platform, and binocular camera, as shown in Figure 1 The control scheme adopts multi-axis integrated control scheme of motion controller and HS servo driver. In addition, the data interaction adopts UDP communication protocol, which is suitable for long-distance transmission in the transmission distance, and the transmission speed and signal fidelity could be greatly improved. The robot device of fracture reduction surgery adopts the parallel Stewart structure, with high control precision. The electric cylinder of six connecting rods adopts a ball screw, which has no clearance and high rigidity, and could realize micro-feed movement and overcome the tension between muscle tissues effectively. The adjustable camera device can adjust the distance to ensure that the area around the fracture is visible in the field of view. For the drive, the servo motor with brake can be protected by power failure, and the servo is controlled by closed-loop feedback.

## 3. Pose Error Modeling of the Fracture Reduction Robot

The structure diagram for FRR is shown in Figure 2. B is the coordinate origin of the fixed platform $B-X_{B}Y_{B}Z_{B}$, P is the coordinate origin of the moving platform $P-X_{P}Y_{P}Z_{P}$, and L is the vector of the fixed platform coordinate origin B pointing to the moving platform coordinate origin P. Setting the coordinates of the i-hinge point of the platform in B as Bi, the coordinates of the i-hinge point of the moving platform in P as Pi, as well as the length of the i-th driving rod as ki, P0,P1⋯P6 as the moving platform hinge point, and B0,B1⋯B6 as the static platform hinge point. The rotation transformation matrix of P with respect to B should be given as R.

In the meantime, *α*, *β*, and *γ* should be the rotation angles of the moving platform coordinate system *p* relative to the fixed platform coordinate system B, which are along the *x*, *y*, and *Z* axes, respectively.
(1)$$R=\left(\begin{array}{cccc} \cos\gamma \cos\beta & -\sin\gamma \cos\alpha +\cos\gamma \sin\beta \sin\alpha & \sin\gamma \sin\alpha +\cos\gamma \sin\beta \cos\alpha & 0\\ \sin\gamma \cos\beta & \cos\gamma \cos\alpha +\sin\gamma \sin\beta \sin\alpha & -\cos\gamma \sin\alpha +\sin\gamma \sin\beta \cos\alpha & 0\\ -\sin\beta & \cos\beta \sin\alpha & \cos\beta \cos\alpha & 0\\ 0 & 0 & 0 & 1 \end{array}\right)$$
where the pose errors of the FRR moving platform can be expressed as:(2)$$\Delta e=\left[\begin{array}{l} dP\\ d\zeta \end{array}\right]$$

In Equation (2) dP = ($\delta P_{x}$, $\delta P_{y}$, $\delta P_{z}$)*^T^* is the position error of the fracture reduction moving platform, and dζ = (δα, δβ, δγ)*^T^* is the attitude error of the fracture reduction moving platform.

For the error modeling of the reset robot, the single-chain closed-loop vector method is adopted. The vector of the six driving rods of FRR from the fixed platform hinge point position to the moving platform hinge point position can be expressed as:(3)$$M_{i}=L+Rp_{i}-b_{i}$$

The length of FRR drive rod is:(4)$$m_{i}=∥M_{i}∥$$

According to Figure 3, the unit vector of each drive rod can be obtained. The vector $u_{i}$ of the six drive rods of FRR points from the hinge point position of the fixed platform to the hinge point position of the moving platform:(5)$$u_{i}=\frac{M_{i}}{∥M_{i}∥}=\frac{M_{i}}{m_{i}}$$

Combining Equations (3) and (5), the *i*-th closed-loop vector could be expressed as:(6)$$u_{i}m_{i}=L+Rp_{i}-b_{i}$$
where, $L=(\begin{array}{ccc} l_{x} & l_{y} & l_{z} \end{array})$.

Differentiate both sides of Equation (6), it could be obtained
(7)$$du_{i}m_{i}+dm_{i}u_{i}=dL+dRp_{i}-db_{i}$$

Multiply both sides of Equation (7) by ${u_{i}}^{T}$ simultaneously:(8)$${u_{i}}^{T}du_{i}m_{i}+{u_{i}}^{T}dm_{i}u_{i}={u_{i}}^{T}dL+{u_{i}}^{T}dRp_{i}{-u_{i}}^{T}db_{i}$$

So, Equation (8) can be reduced to
(9)$$\begin{array}{l} dm_{i}=u_{i}\cdot dL+{\left(Rp_{i}\times u_{i}\right)}^{T}\cdot d\zeta +{u_{i}}^{T}\cdot R\cdot dp_{i}{-u_{i}}^{T}db_{i}\\ =(\begin{array}{cc} u_{i} & {\left(Rp_{i}\times u_{i}\right)}^{T} \end{array})\left(\begin{array}{l} dL\\ d\zeta \end{array}\right)+(\begin{array}{cc} {u_{i}}^{T}\cdot R & {-u_{i}}^{T} \end{array})\left(\begin{array}{l} dp_{i}\\ db_{i} \end{array}\right) \end{array}$$

Additionally, according to the structure of reset robot, there are six drive rods and thereby six closed-loop vectors, so it could be expressed as follows:(10)$$\left(\begin{array}{l} dm_{1}\\ dm_{2}\\ dm_{3}\\ dm_{4}\\ dm_{5}\\ dm_{6} \end{array}\right)=\left(\begin{array}{cc} u_{1} & {\left(Rp_{1}\times u_{1}\right)}^{T}\\ u_{2} & {\left(Rp_{2}\times u_{2}\right)}^{T}\\ u_{3} & {\left(Rp_{3}\times u_{3}\right)}^{T}\\ u_{4} & {\left(Rp_{4}\times u_{4}\right)}^{T}\\ u_{5} & {\left(Rp_{5}\times u_{5}\right)}^{T}\\ u_{6} & {\left(Rp_{6}\times u_{6}\right)}^{T} \end{array}\right)\left(\begin{array}{c} \delta P_{x}\\ \delta P_{y}\\ \delta P_{z}\\ \delta \alpha \\ \delta \beta \\ \delta \gamma \end{array}\right)+\left(\begin{array}{ccccccc} {u_{1}}^{T}\cdot R & {-u_{1}}^{T} & 0 & 0 & \cdots & 0 & 0\\ 0 & 0 & {u_{2}}^{T}\cdot R & {-u_{2}}^{T} & \vdots & 0 & 0\\ 0 & 0 & 0 & 0 & \ddots & 0 & 0\\ 0 & 0 & 0 & 0 & \vdots & 0 & 0\\ 0 & 0 & 0 & 0 & \ddots & 0 & 0\\ 0 & 0 & 0 & 0 & \cdots & {u_{6}}^{T}\cdot R & {-u_{6}}^{T} \end{array}\right)\left(\begin{array}{l} dp_{1}\\ db_{1}\\ \vdots \\ \vdots \\ dp_{6}\\ db_{6} \end{array}\right)$$

Moreover, Equation (10) can be simplified as:(11)$$dm=K_{p}\Delta e+K_{o}dn$$
where: dm = $\left(\begin{array}{l} dm_{1}\\ dm_{2}\\ dm_{3}\\ dm_{4}\\ dm_{5}\\ dm_{6} \end{array}\right)$; $K_{p}$ = $\left(\begin{array}{cc} u_{1} & {\left(Rp_{1}\times u_{1}\right)}^{T}\\ u_{2} & {\left(Rp_{2}\times u_{2}\right)}^{T}\\ u_{3} & {\left(Rp_{3}\times u_{3}\right)}^{T}\\ u_{4} & {\left(Rp_{4}\times u_{4}\right)}^{T}\\ u_{5} & {\left(Rp_{5}\times u_{5}\right)}^{T}\\ u_{6} & {\left(Rp_{6}\times u_{6}\right)}^{T} \end{array}\right)$; Δe = $\left(\begin{array}{c} \delta P_{x}\\ \delta P_{y}\\ \delta P_{z}\\ \delta \alpha \\ \delta \beta \\ \delta \gamma \end{array}\right)$;

$K_{o}$ = $\left(\begin{array}{ccccccc} {u_{1}}^{T}\cdot R & {-u_{1}}^{T} & 0 & 0 & \cdots & 0 & 0\\ 0 & 0 & {u_{2}}^{T}\cdot R & {-u_{2}}^{T} & \vdots & 0 & 0\\ 0 & 0 & 0 & 0 & \ddots & 0 & 0\\ 0 & 0 & 0 & 0 & \vdots & 0 & 0\\ 0 & 0 & 0 & 0 & \ddots & 0 & 0\\ 0 & 0 & 0 & 0 & \cdots & {u_{6}}^{T}\cdot R & {-u_{6}}^{T} \end{array}\right)$; dn = $\left(\begin{array}{l} dp_{1}\\ db_{1}\\ \vdots \\ \vdots \\ dp_{6}\\ db_{6} \end{array}\right)$;

$dp_{i}=\left(\begin{array}{l} dp_{ix}\\ dp_{iy}\\ dp_{iz} \end{array}\right)$; $db_{i}=\left(\begin{array}{l} db_{ix}\\ db_{iy}\\ db_{iz} \end{array}\right)$

Multiplying the two ends of Equation (11) by ${K_{p}}^{-1}$:(12)$$\Delta e={K_{p}}^{-1}dm-K_{o}{K_{p}}^{-1}dn$$

Based on the differential motion of the parallel robot, Δe = δe, dm=δm,dn=δn in Equation (12) can be obtained:(13)$$\delta e={K_{p}}^{-1}\delta m-K_{o}{K_{p}}^{-1}\delta n$$

From Equation (10), we can see that the main factors influencing the positional error of the dynamic platform of the FRR are $dl_{i}$, $db_{i}$ and $dp_{i}$. Meanwhile, Equation (13) is the FRR mechanism error model.

Where: δm—Drive rod error;   $K_{o}$—Vector along the direction of the drive rod;   δe—FRR institutional error;   $K_{p}$—Jacobi Matrix;   δn—Hinge error;   ${K_{p}}^{-1}\delta m$—pose errors due to FRR drive rod length;   $K_{o}{K_{p}}^{-1}\delta n$—Pose error caused by hinge of FRR static platform and moving platform.

Since δn is the hinge error that cannot be compensated for directly, so the hinge error can be converted into a drive rod error that can be measured and controlled, such as: $\delta m^{\prime }=K_{o}\delta n$. Therefore, Equation (13) can be expressed as:(14)$$\delta e={K_{p}}^{-1}(\delta m-\delta m^{\prime })$$

Define $\delta m^{\prime \prime }=\delta m-\delta m^{\prime }$, then Equation (14) can be expressed as
(15)$$\delta e={K_{p}}^{-1}\delta m^{\prime \prime }$$

According to Equation (15), the end pose error of FRR is only related to the error parameters of the drive rod, and the end pose error of FRR can be compensated by adjusting the error parameters of the drive rod.

## 4. Pose Error Analysis Based on Drive Bar Error Variation

Based on the above end-effector error model, the influence of the error variation of the FRR drive rod on the pose error of the moving platform is deeply studied. Figure 4 and Figure 5 shows the variation curve of the end pose error of FRR mechanism with the driving rod error.

Meanwhile, in this sense, the end position and attitude errors of FRR moving platform are changing with other driving rods $\delta m_{3}$, $\delta m_{4}$, $\delta m_{5}$ and $\delta m_{6}$, just in a similar way to that of $\delta m_{1}$ and $\delta m_{2}$. The end-effector position errors $\delta P_{x}$, $\delta P_{y}$, and $\delta P_{z}$ of the moving platform of FRR change with the changing of the driving rod length parameter, and the change range is obvious, while the end-effector attitude errors δα, δβ, and δγ of the moving platform change less than the position errors. It can be seen that the end position error and attitude error of the moving platform do not decrease at the same time when the length error of the same driving rod change. Moreover, when the end position error of the moving platform reaches the minimum, the end attitude error is not the minimum. Therefore, in order to reduce the pose error of the moving platform of the fracture reduction robot as much as possible during the fracture reduction process, and improve the accuracy of FRR during the operation, it is necessary to seek the minimum value of the position error of the moving platform and the attitude error of the moving platform at the same time, so as to complete the correction of positional errors.

## 5. Error Compensation Based on Improved Whale Optimization Algorithm

On the basis of the established FRR attitude error model, the objective function for the correction of attitude error at the end of the moving platform is constructed. The whale optimization algorithm (WOA) is used to solve the multivariate non-linear combination, and when the FRR moving platform attitude error is minimized, the parameters of each drive rod are iterated. This completes the compensation of the FRR end attitude error.

### 5.1. Modified Objective Function

According to FRR end-effector pose error model (Equation (13)), the pose error correction objective function is constructed as follows:(16)$$\eta_{\min}={\left(d{P_{x}}^{2}+d{P_{y}}^{2}+d{P_{z}}^{2}+d\alpha^{2}+d\beta^{2}+d\gamma^{2}\right)}^{0.5}$$

In this equation, the position error and attitude error of the end of the FRR moving platform are fully considered, which are taken as the target of whale optimization algorithm. The optimization algorithm is used to optimize the parameters of the drive rod and compensate the position and attitude error of the end.

### 5.2. Whale Optimization Algorithm

The whale optimization algorithm is divided into three stages: encircling prey, hunting behavior, and searching for prey [27].

#### 5.2.1. Encircling Prey

After identifying the prey, the whale will surround and capture the prey. The target prey is the current optimal solution, and other individuals in the whale herd will update their position according to the current optimal individual, namely:(17)$$D=\left|CX^{*}(t)-X(t)\right|$$
(18)$$X(t+1)=X^{*}(t)-AD$$
where, t is the number of iterations, $X^{*}(t)$ is the position of the optimal solution in the t generation, f is the position of the individual whale, D is the distance between the individual and the prey, and the coefficients A and C represent the convergence factor and the oscillation factor, respectively, which can be obtained by the following formula:(19)$$A=2ar_{1}-a$$
(20)$$C=2r_{2}$$
(21)$$a=2-\frac{2t}{T_{\max}}$$
where, $r_{1}$ and $r_{2}$ are random values between (0,1). The value of a decreases linearly from 2 to 0 as the number of iterations increases. t represents the current number of iterations, and $T_{\max}$ represents the maximum number of iterations.

#### 5.2.2. Hunting Behavior

Based on the hunting behavior of whales, they often approach prey in a spiral pattern. So a mathematical model is established:(22)$$X(t+1)=X^{*}(t)+D_{p}e^{bl}\cos(2\pi l)$$
where, $D_{p}=\left|X^{*}(t)-X(t)\right|$ represents the distance from the individual whale to the target prey, $X^{*}(t)$ represents the best position at present, *b* represents a constant spiral shape, and l takes a random number in (−1,1). In the process of whale predation, whales have a shrinkage encirclement mechanism and a spiral model to update the whale’s position during predation, with probability of 50%. The mathematical model is as follows:(23)$$X(t+1)\left\{ \begin{array}{l} X^{*}(t)-AD,p<0.5\\ X(t)=X^{*}(t)+D_{p}e^{bl}\cos(2\pi l),p\geq 0.5 \end{array}\right.$$

A is a random value within [−a,a], and when A is within [−1,1], the whale attacks its prey.

#### 5.2.3. Searching for Prey

When A≥1, individuals in the whale herd will randomly select prey according to each other’s positions to improve the global search ability of the algorithm and establish a mathematical model:(24)$$D=\left|CX_{rand}-X(t)\right|$$
(25)$$X(t+1)=X_{rand}-AD$$
where $X_{rand}$ is the random whale position.

As shown in Figure 6, the optimization process of WOA is conducted. For the compensation of FRR end poses, the value range $m_{i\min}\leq m_{i}\leq m_{i\max}$ of the rod length of the drive rod is used as the search space. Moreover, the whale optimization algorithm is used to iterate continuously to obtain the parameter error of each rod length when the minimum error of the mechanism error of the fracture reduction robot is obtained, so as to complete the error compensation.

Meanwhile, the realization process of FRR error compensation is shown in Figure 7. According to the given FRR reduction process task, the pose error model established by the reduction robot according to Equation (13) and the pose error correction objective function of the fracture reduction robot established by Equation (16) were used to obtain the drive rod adjustment DL based on the optimization algorithm, which was added to the rod length obtained through inverse kinematics solution. To correct the pose error.

In addition, Figure 8 shows the change of the end-effector pose errors with the increase in the number of iterations, with the increase in the number of iterations, the degree of fitness gradually decreases and finally stabilizes. For Min = 1.423 × 10^−3^, the correction effect is relatively obvious.

Moreover, Table 1 shows the comparison of poses errors of the mobile platform of the fracture reduction robot before and after the optimization of the length error parameters of the drive rod:

### 5.3. Improved Whale Optimization Algorithm Based on Differential Evolution

The local development ability of WOA is stronger than the global, which leads to the poor global search ability of WOA in the later stage, and easy to fall into the local optimum. In view of the above shortcomings, this paper proposes to adopt Cauchy opposition-based learning and DE to improve the speed and accuracy.

#### 5.3.1. Opposition-Based Learning

It is given that a point in D-dimensional space is $X=x(x_{1},x_{2},x_{3},\ldots x_{n})$, where i=1,2,3,…,n, $n_{i}$ are i-dimensional minima and $m_{i}$ is a maximum. The opposite value of point *X* is:(26)$$\bar{X}=\bar{x}({\bar{x}}_{1},{\bar{x}}_{2},{\bar{x}}_{3},\ldots {\bar{x}}_{n})$$
where: ${\bar{x}}_{i}=m_{i}+n_{i}-x_{i}$

On the basis of opposition-based learning [28,29], Cauchy’s idea of opposition-based learning is introduced. This paper uses opposition-based learning to generate the initial population and improve convergence speed of WOA. The Cauchy opposite value could be defined: setting a point in D-dimensional space is $X=x(x_{1},x_{2},x_{3},\ldots x_{n})$, where i=1,2,3,…,n, $n_{i}$ are i-dimensional minima and $m_{i}$ is a maximum. The Cauchy opposite value of point *X* is:(27)$$\bar{C}\bar{X}=rand(\frac{m_{i}+n_{i}}{2},m_{i}+n_{i}-x_{i})$$

#### 5.3.2. Differential Evolution Algorithm

Differential evolution algorithm mainly consists of mutation operation, crossover operation and selection operation. The main control parameters of a differential evolution algorithm include population size (NP), scaling factor (F) and crossover probability (CR).

(1) Initial population
(28)$$X_{ij}(0)=X_{ij,\min}+rand(0,1)(X_{ij,\max}-X_{ij,\min})$$
where: i=1,2,…,N, N is population number, $X_{ij,\max},X_{ij,\min}$ are the maximum and minimum values of variable j of individual i.

(2) Mutation operation

To mutate an individual in a population:(29)$$X_{i}(t+1)=X_{1}(t)+F*(X_{2}(t)-X_{3}(t))$$
where $X_{1}(t),X_{2}(t),X_{3}(t)$ are vectors of randomly selected, two unequal individual vectors; t is the number of evolutions; F is the scaling factor, which affects the global optimization ability of the algorithm, F∈[0,2].

(3) Crossover operation

After mutation, individual $X_{ij}(t+1)$ and the original individual $X_{ij}(t)$ in the population exchange some elements to generate trial individuals, thereby expanding the size of the population:(30)$$V_{i,j}(t+1)=\{\begin{array}{cc} X_{ij}(t+1) & rand1\leq CR\;or\;j=rand2\\ X_{ij}(t) & rand1>CR\;or\;j\neq rand2 \end{array}$$
where: $V_{i,j}(t+1)$ A is the new individual after crossover;

rand1 is a random number between [0,1];

rand2 is a random number between [0,D], D is the dimension of the variable, and

CR is the crossover probability.

(4) Selection operation

Comparing the new individual $V_{i,j}(t+1)$ obtained by the crossover operation with the experimental individual $X_{ij}(t)$, if the fitness of the new individual $V_{i,j}(t+1)$ is worse than that of the experimental individual $X_{ij}(t)$, the new individual will replace the experimental individual, and enter the offspring [30]. Even if that the fitness of the new individual is not as good as that of the experimental individual, then the experimental individual will enter the offspring:(31)$$X_{i}(t+1)=\{\begin{array}{cc} V_{i}(t+1) & f(V_{i}(t+1))<f(X_{i}(t))\\ X_{i}(t) & f(V_{i}(t+1)\geq f(X_{i}(t)) \end{array}$$
where f is the fitness function.

Finally, this paper simulates the differential evolution algorithm to improve the whale optimization algorithm, improve the population richness, and improve the optimization accuracy. Cauchy reverse learning was used to generate the initial population and the reverse population of the current population when the algorithm evolved, and the crossover and selection strategy of DE algorithm was introduced into the WOA algorithm to improve the diversity of the population during the evolution. The whale position is regarded as an individual, and the location update is completed through the difference strategy. The fitness value of individual whale is calculated to determine the optimal individual, and the spiral update and contract-enveloping methods are used to update the whale position. Finally, the selection of the optimal individual is selected according to the fitness value. The whale optimization algorithm with DE algorithm is shown in Figure 9. By introducing reverse learning and the DE algorithm, the whale optimization algorithm has better global optimization ability, faster search speed, and higher iteration accuracy.

## 6. Results and Discussions

### 6.1. Comparison of FRR End Pose Optimization Algorithms

Based on the FRR experimental platform, this paper completed the compensation of the end poses error of the FRR by iterating the parameters of the drive rod with the algorithm. In this section, the improved WOA with Cauchy opposition-based learning and differential evolution algorithm is compared with the general WOA. The simulation results are shown in Figure 10. It can be seen that the improved WOA has a significant improvement in the optimization speed and accuracy. Furthermore, in Table 2, optimization and convergence precision of WOA with Cauchy opposition-based learning and differential evolution algorithm and WOA only are shown. It can be seen that detail simulation experiment illustrated the improved optimization algorithm can effectively overcome global convergence ability weak faults of the WOA, and both the convergence speed and optimization precision compared had been improved. It is confirmed that the optimization accuracy is improved by 50.74% and the convergence speed is improved by 22.62%.

Meanwhile, in order to further verify that the improved WOA has a great improvement in optimization accuracy, this paper compares the end pose error of the FRR compensated by the PSO algorithm and AC algorithm. The result curve is displayed in Figure 11 and Table 3.

It can be inferred from Figure 11 and Table 3 that, in contrast to the whale optimization algorithm, the particle swarm algorithm, the ant colony algorithm, and the general optimization algorithm, the improved WOA has a better effect on the fracture end of the robot posture error compensation than other algorithms, and the convergence speed and optimization accuracy are greatly improved.

As the whale optimization algorithm is relatively similar to the grey wolf optimization algorithm (GWO), the applied improved whale optimization algorithm is compared with the improved grey wolf optimization algorithm. Although the two algorithms are similar, the simulation results for the effect of optimizing the robot end position error in this paper show that the improved whale optimization algorithm proposed in this paper has a faster convergence rate and higher accuracy than the grey wolf optimization algorithm. The results of the GWO and improved GWO iterations and the optimization results are shown in the Figure 12 and Table 4.

### 6.2. FRR Reset Simulation Experiment

In parallel to the algorithm simulation, as shown in Figure 13, the MATLAB Simulink toolbox and SimMechanics were used to establish the FRR’ 3D models, and conducted simulation experiments.

A set of desired trajectories is shown in Table 5 and the simulation model built by Simmechanics was used to simulate the resetting process of the FRR model as shown in Figure 14. Meanwhile, the resetting process of the FRR prototype is shown in Figure 15. The desired length variation curve and the actual variation curve of each branch chain along the desired trajectory obtained from the inverse dynamics model control are shown in Figure 16. Correspondingly, Figure 16a shows the desired length change curve of each branch chain and Figure 16b shows the actual length change curve of each branch chain under kinetic control. It can be seen that each drive rod reaches the desired attitude within 6 s according to the set desired attitude. The motion of each drive rod is generally consistent with the variation pattern obtained from the inverse attitude solution, which proves the correctness of the FRR reset simulation for the given desired attitude.

Furthermore, Figure 17 shows the error between the actual trajectory and the expected trajectory of the moving platform. It can be seen that each branch chain followed the expected variation accurately and was able to reach the expected attitude in exactly 6 s. In Figure 17a, the error between the actual motion trajectory and the expected trajectory before algorithm optimization is shown, as well as after algorithm optimization, in Figure 17b. In Figure 18, the difference between the error before and after optimization in the *x*, *y*, and *z* axis directions are shown, and it can be seen that the effect of the algorithmic optimization is relatively obvious. The errors along both the *x* and *z* axes are significantly reduced. The 100 data points of the pre- and post-optimization error curves were also selected at equal time intervals and the corresponding data series before and after extraction of the optimization were analyzed for differences in means using a paired-samples *t*-test. The results of the paired-samples *t*-test are shown in Table 6, with a significant reduction in error after optimization compared to before optimization (*x*-axis: *t* = 64.923, *p* < 0.01; *z*-axis: *t* = 33.560, *p* < 0.01), demonstrating the effectiveness of the optimization algorithm in reducing error compensation during FRR resetting.

## 7. Conclusions

This paper establishes the end position error model of the FRR dynamic platform and finds that the end position error is related to the drive rod parameters of the fracture repositioning robot through the error model analysis. The influence of the parameters of the drive rod on the end position and posture error is also analyzed. The fracture repositioning robot’s posture error correction objective function is constructed. To further improve the optimization accuracy, an improved whale optimization algorithm is utilized in this paper, and compared with the whale optimization algorithm (WOA), the iteration accuracy of CRLWOA-DE is improved by 50.74% and the optimization speed is increased by 22.62%. At the same time, CRLWOA-DE proved to be more accurate compared to particle swarm optimization (PSO) and ant colony optimization (ACO), as well as the grey wolf optimization algorithm, which is similar to the whale optimization algorithm. In addition, a reconstruction of the fracture-resetting robot was carried out using SimMechanics in MATLAB software to verify the actual trajectory of the mobile platform optimized by CRLWOA-DE. The errors along the *x* and *z* axes were significantly reduced from the expected trajectory, and the differences in the mean values of the corresponding data series before and after extraction of the optimization were analyzed using paired samples *t*-tests. The errors were significantly reduced after optimization compared to the pre-optimization period (*x*-axis: *t* = 64.923, *p* < 0.01; *z*-axis: *t* = 33.560, *p* < 0.01. It was demonstrated that the CRLWOA-DE method can be used to reduce postural errors in fracture-repositioning robots, which has implications for bone fracture and deformity correction. Although this study has yielded some meaningful results, it also has some limitations. The current experiment only considered the skeletal reduction process, but during the clinical process, changes in muscle shape can affect the error during skeletal docking. Next, the compensation of posture errors at the end of the moving platform for the robot with muscles in the reset process will be further studied.

## Figures and Tables

**Figure 1 micromachines-13-01186-f001:**
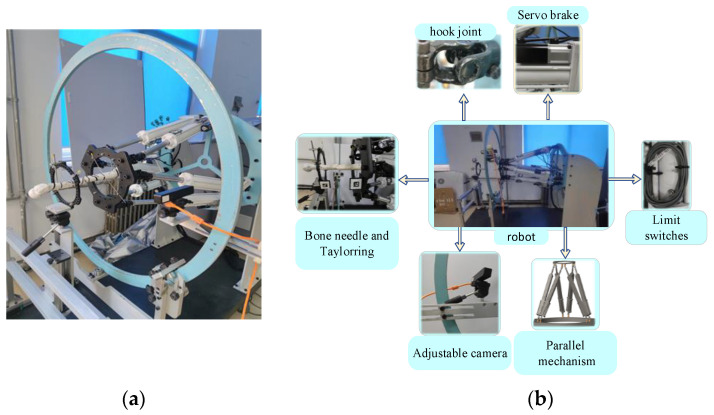
Structural design of the fracture reduction robot: (**a**) overall view; (**b**) key parts view.

**Figure 2 micromachines-13-01186-f002:**
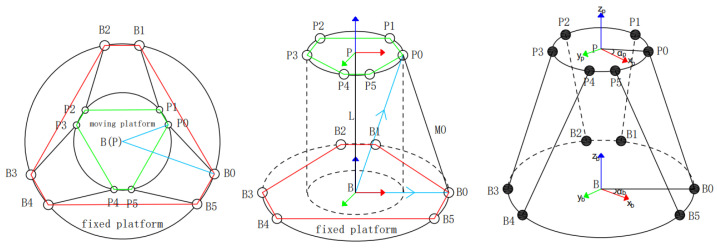
Structure diagram of FRR.

**Figure 3 micromachines-13-01186-f003:**
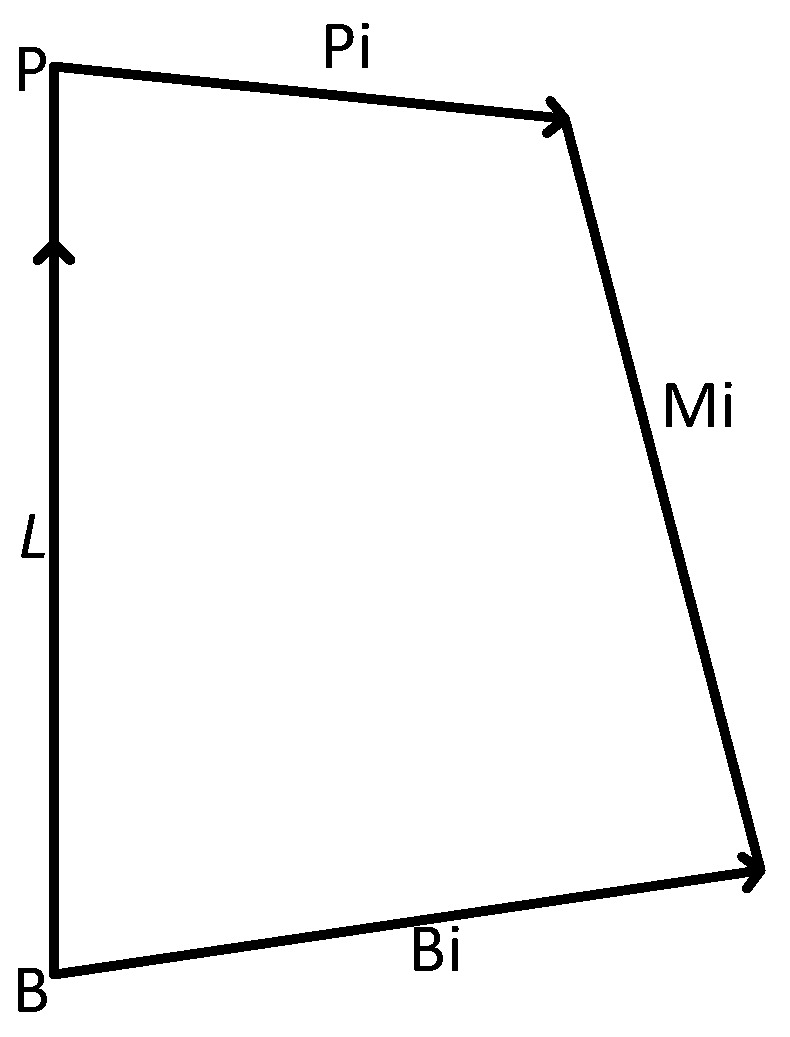
Closed-loop vector diagram of a single branch chain.

**Figure 4 micromachines-13-01186-f004:**
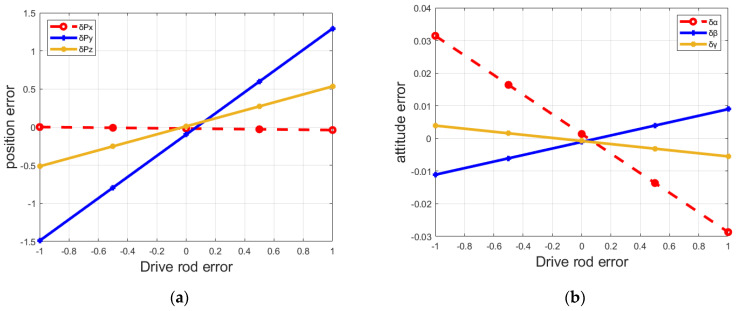
Influence of drive rod 1 error on FRR end-effector pose error: (**a**) drive rod 1 error on positional error; (**b**) drive rod 1 error on attitude error.

**Figure 5 micromachines-13-01186-f005:**
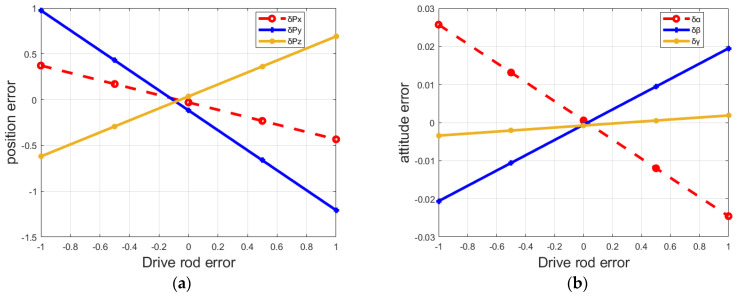
Influence of drive rod 2 error on FRR end-effector pose error: (**a**) drive rod 2 error on positional error; (**b**) drive rod 2 error on attitude error.

**Figure 6 micromachines-13-01186-f006:**
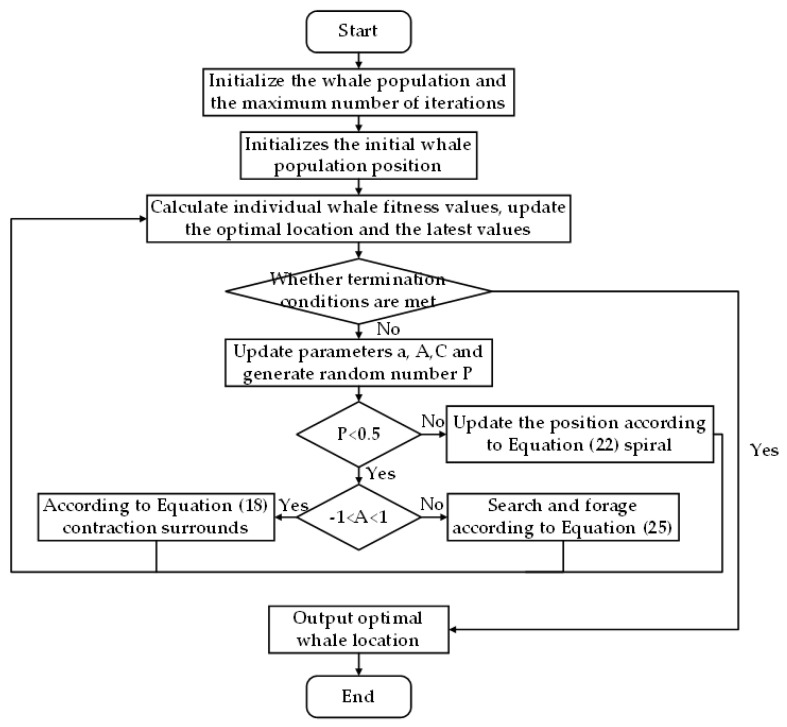
The whale optimization algorithm optimizes the process.

**Figure 7 micromachines-13-01186-f007:**
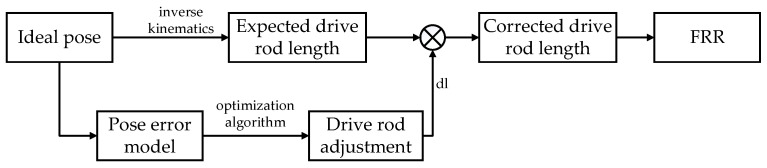
FRR error compensation block diagram.

**Figure 8 micromachines-13-01186-f008:**
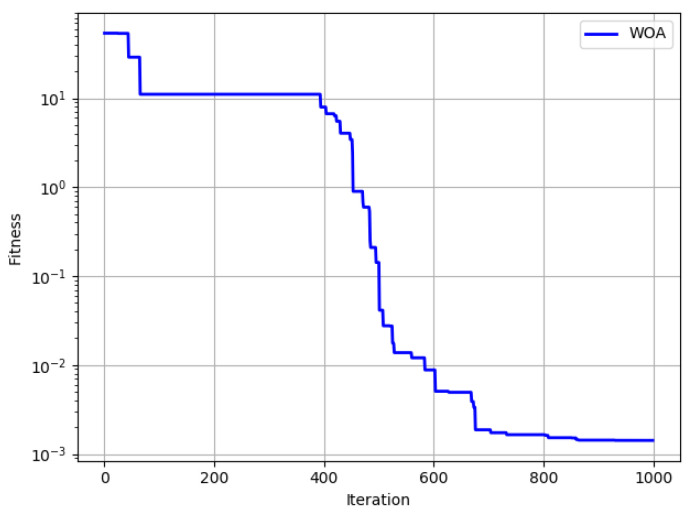
WOA corrected end position error curve.

**Figure 9 micromachines-13-01186-f009:**
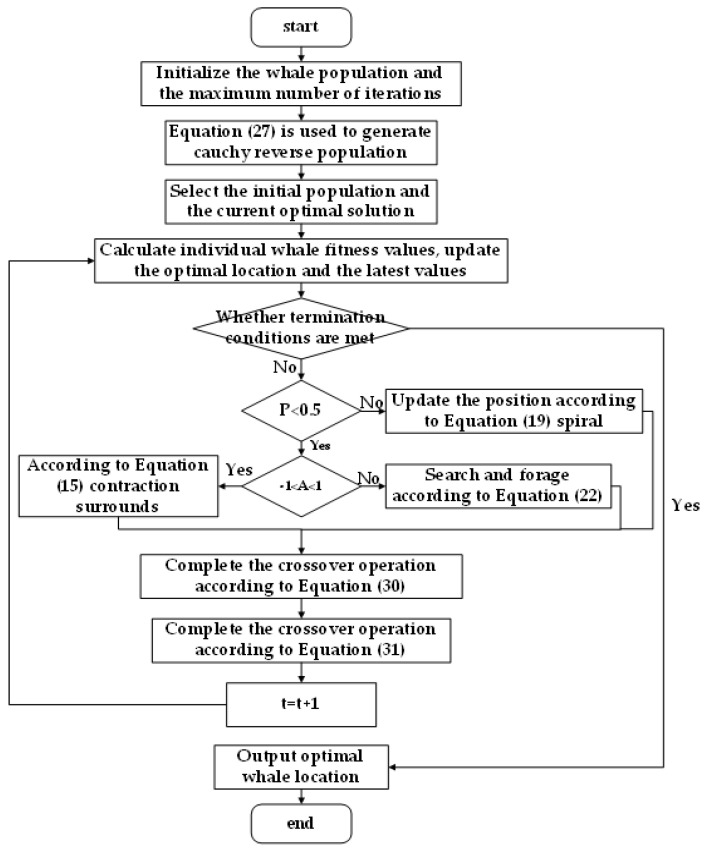
Whale optimization algorithm optimization flow chart with the DE algorithm.

**Figure 10 micromachines-13-01186-f010:**
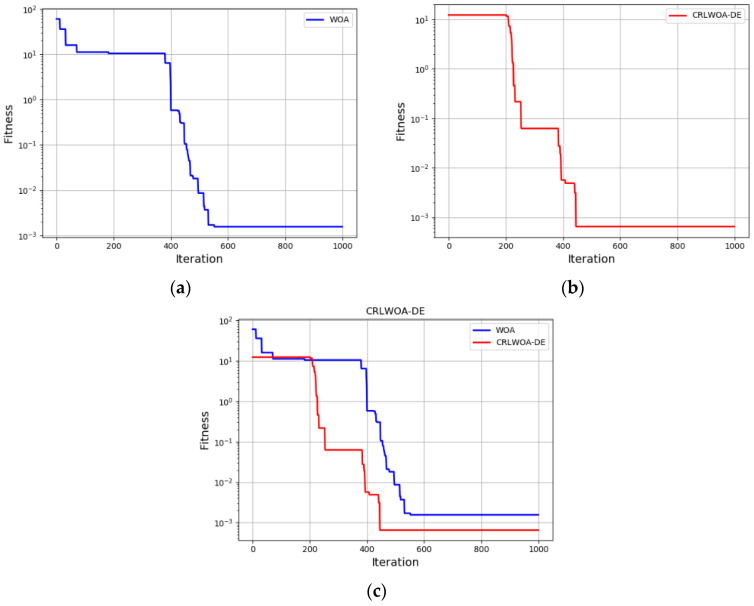
(**a**) WOA optimization curve; (**b**) CRLWOA-DE optimization curve; (**c**) convergence curves of WOA and CRLWOA-DE.

**Figure 11 micromachines-13-01186-f011:**
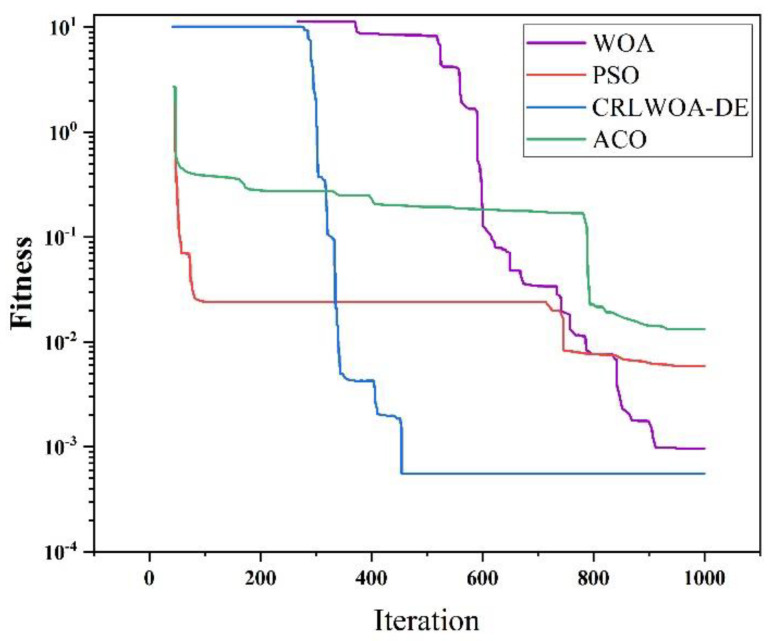
ACO, WOA, CRLWOA-DE, and PSO algorithm iteration results.

**Figure 12 micromachines-13-01186-f012:**
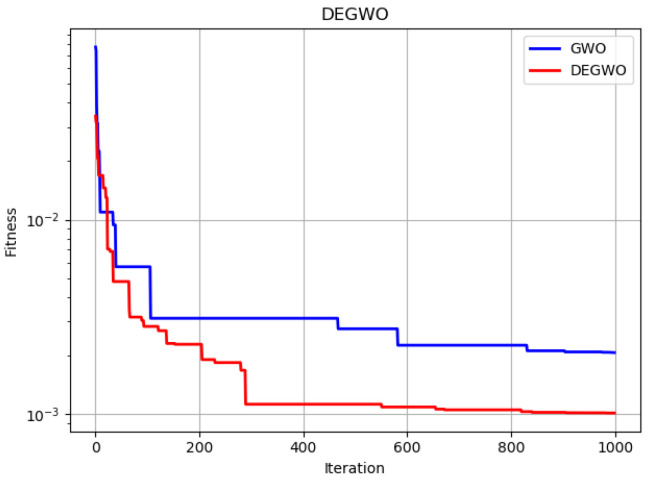
Results of iterations of the GWO and improved GWO algorithms.

**Figure 13 micromachines-13-01186-f013:**
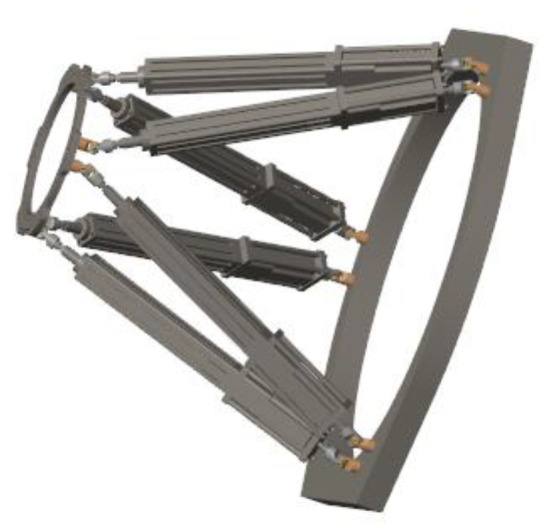
Robot model constructed by SimMechanics.

**Figure 14 micromachines-13-01186-f014:**
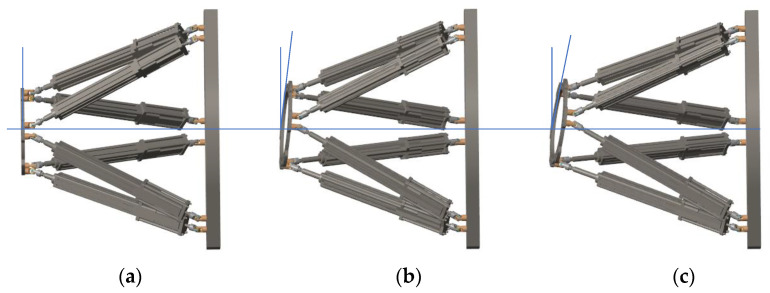
(**a**–**c**) are the process of FRR 3D model.

**Figure 15 micromachines-13-01186-f015:**
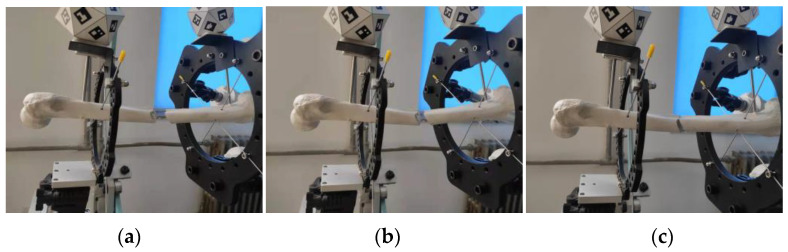
(**a**–**c**) are the process of FRR prototype.

**Figure 16 micromachines-13-01186-f016:**
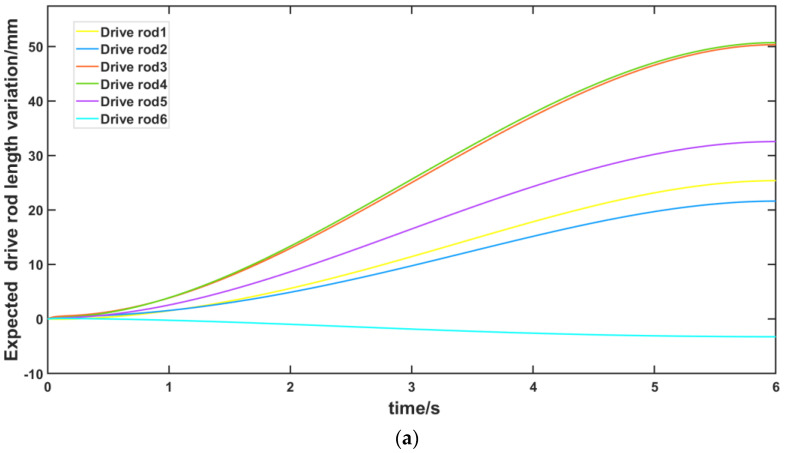
Length change curve of driving rod in moving platform: (**a**) expected drive rod length change curve with time; (**b**) actual driving rod length variation curve with time.

**Figure 17 micromachines-13-01186-f017:**
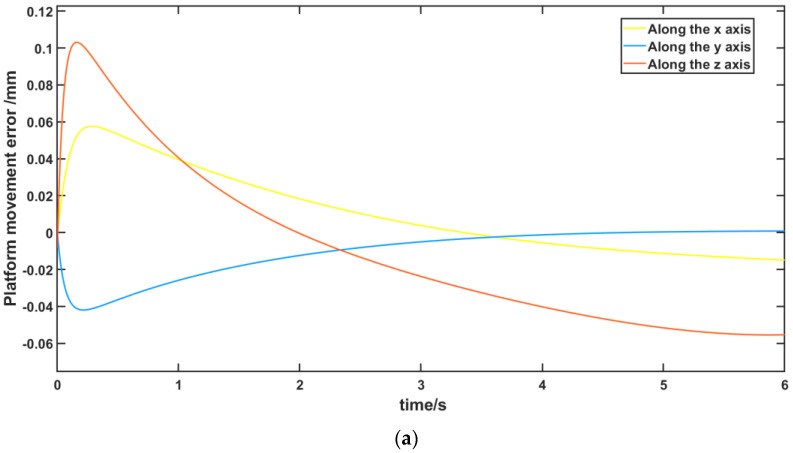
Comparison of the error between the actual motion trajectory and the expected motion trajectory before and after mobile platform optimization: (**a**) comparison of the error between the actual motion trajectory and the expected motion trajectory before mobile platform optimization; (**b**) comparison of the error between the actual motion trajectory and the expected motion trajectory after mobile platform optimization.

**Figure 18 micromachines-13-01186-f018:**
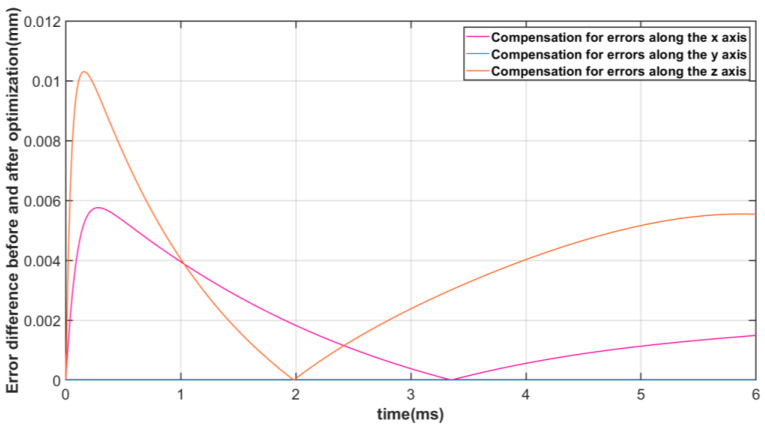
Error optimization effect of actual and expected motion trajectory of mobile platform.

**Table 1 micromachines-13-01186-t001:** Comparison of pose errors before and after WOA algorithm optimization.

	$\delta P_{x}$	$\delta P_{y}$	$\delta P_{z}$	δα	δβ	δγ
Before optimization	−0.0538	−0.0453	0.0027	−0.0001	−0.0016	−0.0008
After optimization	0.3215 × 10^−3^	0.5986 × 10^−3^	−0.494 × 10^−3^	−0.00134	0.000219	−0.000423

**Table 2 micromachines-13-01186-t002:** Comparison of optimization results between whale optimization algorithm and improved whale optimization algorithm.

The Simulation Serial Number	WOA		CRLWOA-DE	
	Optimizing Accuracy	Number of Convergences Required	Optimizing Accuracy	Number of Convergences Required
1	0.00147976	850	0.00060458	560
2	0.00186629	900	0.00108461	520
3	0.00120779	525	0.00050738	500
4	0.00142077	655	0.00050963	625
5	0.00276931	935	0.00053662	575

**Table 3 micromachines-13-01186-t003:** ACO, WOA, CRLWOA-DE, and PSO algorithm optimization results.

	$\delta p_{x}$	$\delta p_{y}$	$\delta p_{z}$	δα	δβ	δγ
Particle swarm optimization	0.5156 × 10^−3^	0.0710 × 10^−3^	0.5436 × 10^−3^	−0.8482 × 10^−3^	−0.4861 × 10^−3^	−0.0568 × 10^−3^
Ant colony algorithm	0.90 × 10^−3^	3.90 × 10^−3^	−3.017 × 10^−3^	0.5776 × 10^−3^	−0.7150 × 10^−3^	3.173 × 10^−3^
Whale optimization algorithm	−0.2522 × 10^−3^	0.0315 × 10^−3^	−0.1767 × 10^−3^	0.2371 × 10^−3^	0.4069 × 10^−3^	−0.3435 × 10^−3^
Improved whale optimization algorithm	0.0023 × 10^−3^	−0.0110 × 10^−3^	0.0261 × 10^−3^	−0.1688 × 10^−3^	−0.2035 × 10^−3^	0.0266 × 10^−3^

**Table 4 micromachines-13-01186-t004:** Improved GWO optimization results.

	$\delta p_{x}$	$\delta p_{y}$	$\delta p_{z}$	δα	δβ	δγ
Improved grey Wolf optimization algorithm	−0.0152 × 10^−3^	−0.0987 × 10^−3^	−0.0674 × 10^−3^	−0.00172	0.00051	−0.00009

**Table 5 micromachines-13-01186-t005:** Initial pose and expected pose of FRR moving platform.

	*x*/mm	*y*/mm	*z*/mm	α	β	γ
Initial pose	0	0	0	0	0	0
Desired pose	30	10	30	7	8	9

**Table 6 micromachines-13-01186-t006:** Paired samples *t*-test.

Paired Sample Test								
	Paired Difference					*t*	DOF	Sig.
	Average	Standard Deviation	Standard Mean Error	Difference 95% Confidence Interval				
				Lower Limit	Upper Limit			
Before and after optimization along the *x*-axis	0.005198529	0.000796711	0.00008007253	0.005039628038	0.005357430734	64.923	98	0.000
Before and after optimization along the *z*-axis	0.007569028	0.002244068	0.00022553734	0.007121457234	0.008016600279	33.560	98	0.000

## Data Availability

Not applicable.

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
