# Peer review of "Study on the Modeling and Compensation Method of Pose Error Analysis for the Fracture Reduction Robot"

_micromachines, 2022, doi:10.3390/mi13081186_

Round 1
Reviewer 1 Report
The authors of the paper describe their proposed approach for Study on the Modeling and Compensation Method of Pose Error Analysis for the Fracture Reduction Robot. The topic is interesting and with possible future real applicability. However, the paper needs several improvements and corrections:
1) the main contribution and originality should be explained in more detail, is the proposal of modified whale optimization with differential evolution? or the appplication to robotics?
2) the motivation of the approach with metaheuristic optimization needs further clarification, why the whale combined with DE? why not other metaheuristic?
3) discussion of related work in metaheuristic optimization for robotics should be expanded with more recent work
4) Minor grammar and syntax issues need correction to enhance readability of the paper
5) more simulation results and formal comparison of results are needed, a comparison with grey wolf (GWO) would be interesting, as whale optimization is very similar to GWO
6) the conclusions should be extended with more future work
7) there are several typos in the paper, including some of them in the references
8) More references to recent related papers could be included, like for example:
Overtaking Uncertainty With Evolutionary TORCS Controllers: Combining BLX With Decreasing α Operator and Grand Prix Selection
Optimal Setting of Membership Functions for Interval Type-2 Fuzzy Tracking Controllers Using a Shark Smell Metaheuristic Algorithm
Optimal Fuzzy Controller Design for Autonomous Robot Path Tracking Using Population-Based Metaheuristics
Reviewer 2 Report
This paper based on self-developed parallel mechanism fracture reduction robot (FRR), a novel method to reduce the pose errors of FRR is proposed. Firstly, this paper analyzed the pose errors, and built the model of pose errors. Secondly, mechanism errors of FRR were converted into drive bar parameter’s errors, and the influence of each drive bar parameters on the robot pose errors were analyzed. Thirdly, combining with Cauchy opposition-based learning and differential evolution algorithm, an improved whale optimization algorithm (CRLWOA-DE) is proposed to compensate the end-effector’s pose errors. The paper could be of interest with respect to the topic of the journal. However, I have some comments.
1. It is not clear which is the novelty of the present work.
2. Which open issue that authors are trying to solve.
3. A review of state of the art is completely missing. Authors shall present a review of previous work on the topic. I suggest adding a Related Works section where the authors state clearly which is the novelty wrt to previous works and which issues are trying to solve.
4. The authors shall present briefly the same concepts in the introduction which is now too generic.
5. The abstract and conclusion need improvement as per international standards.
Reviewer 3 Report
1. Metaheuristic belongs to stochastic optimization method and is mainly driven by the random streams utilized in the search mechanism. A recently novel idea of controlling the pseudo random streams in the stochastic optimization method should be mentioned and noticed in the introduction section. The following work is suggested to be added in the manuscript.
[1] Stochastic Stability Analysis of Particle Swarm Optimization with Pseudo Random Number Assignment Strategy, European Journal of Operational Research, 2022. https://doi.org/10.1016/j.ejor.2022.06.009
2. In the current version, the simulated results are provided without quality measures (standard error/deviation). Please add the quality measures for the corresponding estimates.
3. The statistical tests (parametric or nonparametric testing method) should be performed for the performance comparisons, and then conclude the results based on the statistical tests.
Round 2
Reviewer 1 Report
The authors have addressed all my concerns and the paper can be accepted.
Reviewer 3 Report
The authors seem to have done their best to address the reviewers’ comments. I am recommending this paper for publication to acknowledge the very commendable attempts of the authors to modify and improve it.